# Comparison of patient perceptions of primary care quality across healthcare facilities in Korea: A cross-sectional study

Yongjung Cho[1], Heeyoung Chung[1], Hyundeok Joo[1], Hyung Jun Park[2,3,4], Hee-Kyung Joh[4,5,6]*, Ji Won Kim[1,7], Jong-Koo Lee[6,8]

**1** Seoul National University College of Medicine, Seoul, Republic of Korea, **2** Department of Family Medicine, SMG-SNU Boramae Medical Center, Seoul, Republic of Korea, **3** Clinical Medical Sciences, College of Medicine, Seoul National University, Seoul, Republic of Korea, **4** Department of Family Medicine, Seoul National University Health Service Center, Seoul, Republic of Korea, **5** Department of Medicine, Seoul National University College of Medicine, Seoul, Republic of Korea, **6** Department of Family Medicine, Seoul National University Hospital, Seoul, Republic of Korea, **7** Department of Internal Medicine, SMG-SNU Boramae Medical Center, Seoul, Republic of Korea, **8** Center for Healthy Society and Education, Seoul National University College of Medicine, Seoul, Republic of Korea

\* hkjoh@snu.ac.kr

**Data Availability Statement:** \* All relevant data are within the paper and its Supporting Information files.

## Abstract

Primary care is not well established in Korea despite its importance in population health. To reinforce the primary care system, understanding the public view of primary care will be essential. We aimed to compare the public perception of primary care qualities across types of healthcare facilities. We conducted a cross-sectional, web-based survey at a university in Seoul, South Korea, from October 2018 to February 2019. Using the Korean Primary Care Assessment Tool (K-PCAT), participants assessed the qualities of primary care services provided by the university health service (a university-based, patient-centered primary care model), community clinics, and hospitals. We compared K-PCAT scores across facilities and evaluated the factors associated with the differences using general linear models. A total of 5,748 responses were analyzed. K-PCAT total scores were highest for the university health service (61.0 ± 15.9) and lowest for hospitals (48.1 ± 14.5), with significant differences between facilities (*P* < .001). The university health service received the highest scores for first contact, comprehensiveness, personalized care, and family/community orientation; community clinics for continuity of care; and hospitals for care coordination and trust/satisfaction. Primary care facilities were rated higher than hospitals by individuals in good health, with low income levels, using ambulatory care more frequently, and spending less on medical expenses. In conclusion, the user-perceived primary care quality was higher for community-based primary care facilities than hospitals. The highest score was for the university health service, suggesting that setting-based, patient-centered primary care would be an effective model for restructuring the primary care system in Korea.

**Funding:** The authors received no specific funding for this work.

**Competing interests:** The authors have declared that no competing interests exist.

**Abbreviations:** β, regression coefficient; K-PCAT, Korean Primary Care Assessment Tool; OECD, Organisation for Economic Cooperation and Development; SD, standard deviation; SE, standard error; WHO, World Health Organization.

# Introduction

The sustainability of health systems is at stake worldwide due to rising healthcare costs.[1,2] The World Health Organization (WHO) has estimated that 20% to 40% of healthcare spending is wasted through inefficient use of healthcare.[3] To effectively deal with the global challenges, the WHO recommended that all countries focus on strengthening their primary care systems.[2] Countries with a strong primary care sector have better population health outcomes, care qualities, and user satisfaction at lower costs and with a more equitable distribution of resources than countries with specialist-oriented systems.[4–10] Despite the evidence for primary care, however, resource allocation in most countries is still skewed toward hospitals and specialist care.[6,11–13]

The Korean health system has developed very rapidly in a short period, especially after the introduction of National Health Insurance in 1989—a universal coverage system compulsory for all citizens. The nation's efforts have been heavily focused on hospital-centered specialty care to meet growing demands of the population.[14–17] In contrast, the primary care sector has not been well established and has continued to deteriorate.[18] The strength of the Korean primary care system was reported as the weakest among the Organisation for Economic Cooperation and Development (OECD) countries.[19] Many unique features of the Korean context have hindered the establishment of a primary care system, including the nominal patient referral system without a gatekeeping function,[17,20–23] fee-for-service schedule,[18] overproduction of specialists and sub-specialties,[20,24] and private sector dominance.[22] Hospitals in Korea operate large-scale outpatient units, providing primary care as well as secondary/tertiary care services.[21] Due to the absence of functional gatekeeping (virtually equivalent to direct access to hospitals),[23,24] hospital ambulatory care has been over-utilized, and many hospitals in turn have expanded their outpatient units.[21] Furthermore, their intensive use of expensive technology and over-provision of treatment services have burdened the National Health Insurance system,[6,16,17,25] undermining efficiencies in health expenditure and equity in resource allocation.[16] Previous studies estimated that at least 15% to 33% of hospital outpatient visits were deliverable in primary care facilities,[24,26] and 29% of the national expenditure on ambulatory care was for the hospital sector.[24] Consequently, Korea has experienced one of the highest rates of health expenditure increases among OECD countries.[16]

To overcome these problems, reforming health systems to make better use of resources is crucial.[25] National policy changes are urgently needed to shift focus from a specialty- and curative-care orientation to a more cost-effective health system based on high-quality primary care.[20,21,27] As the first step to reinforce the primary care system, understanding public perception of primary care will be essential.[27] Many studies proved the importance of the user views and satisfaction: patients' perception of good quality of care was associated with their compliance, continuous patient-physician relationships, and better health outcomes.[7,28] In the unique context of Korea, where patients can directly access any type of care facilities, understanding and incorporating the public views and perceptions will be more important. Nonetheless, only few studies have evaluated the public perceptions of primary care quality in Korea. We aimed to compare user perceptions of primary care qualities across different types of healthcare facilities. Our findings would provide useful insights into future directions to restructure the health systems in Korea and countries with similar circumstances.

# Methods

## Participants and procedure

We conducted a cross-sectional, web-based survey from October 2018 to February 2019 at Seoul National University, the largest public university in South Korea. Participants

were recruited from the entire students, staff, and faculty of the university (n = 35,026). We sent potential participants invitation e-mails explaining the purpose and an outline of the survey. A web link to the survey was posted on the university's intranet, websites, and social media. We consecutively enrolled participants who voluntarily responded to and self-administered the survey. Inclusion criteria were those aged ≥18 years and who agreed to answer the survey. Exclusion criteria were individuals who did not give consent and those with missing data on more than 80% of the main outcome measurement. The study was approved by the Institutional Review Board of Seoul National University College of Medicine/Seoul National University Hospital (Seoul, South Korea; IRB number, H-1807-062-957). All participants provided informed consent online before participation in the survey.

## Measures

Participants were asked to compare, based on their past experiences, the qualities of primary care provided by 3 different types of healthcare facilities: the university health service, community clinics, and hospitals. A university health service is a primary care model on campus dealing with medical care and health promotion of university students and employees. The university health service in this university implements a wide range of activities: primary medical consultation for acute and chronic diseases, mental health care, communicable disease prevention, immunizations, health screening, and health education. Family physicians play the central role in these activities, partly supported by some specialty care.

Participants' perceptions of care quality was assessed using the Korean Primary Care Assessment Tool (K-PCAT),[29] which measures key attributes of primary care related to effective care organization and delivery at the population level.[30] In the current study, the K-PCAT was modified into 7 domains with 18 subordinate questions (S1 and S2 Files). The 7 domains consisted of: first contact (accessibility of care at the time of need and its utilization towards solving a new health problem), comprehensiveness (a range of services encompassing common problems in the population), continuity (the longitudinal use of a regular source of care over time, focusing on the long-term health of a person, rather than the short-term duration of the disease), coordination (the role of coordinating other specialist services or resources that patients may need), personalized care (a holistic approach that considers physical, mental, and spiritual well-being), family/community orientation (the inclusion of family health concerns in decision-making and the provider's knowledge of community health needs), and trust/satisfaction.[6,27,30–32] Responses were obtained using a 5-point Likert scale (5 = strongly agree, 4 = agree, 3 = neutral, 2 = disagree, 1 = strongly disagree). Subtotal scores for each domain were yielded by adding up the subordinate item scores. A total K-PCAT score was calculated by summing the subtotal scores of the 7 domains.

As correlated variables, we obtained information on various sociodemographic characteristics and healthcare utilization. Sociodemographic variables included age, sex, job (student/staff/faculty), income level (low/middle/high), self-perceived health (good/fair/poor), presence of medical doctor in one's family (yes/no), and comorbid medical conditions. Comorbidity was self-reported and categorized into 3 groups: none, only acute conditions, and chronic conditions. Chronic condition was defined as the presence of any of the following: hypertension, diabetes, hyperlipidemia, heart disease, hyperuricemia, chronic viral hepatitis, arthritis, cancer, depression, and anxiety disorder. Variables on healthcare utilization included having a regular doctor (yes/no), the number of ambulatory care uses per year (0–3, 4–6, 7–12, ≥13), and medical expenses per year (out-of-pocket payment only). We asked if anyone in their family regularly visited a doctor for chronic conditions (yes/no).

## Statistical analysis

Characteristics of the participants were summarized as numbers and percentages for categorical variables and means ± standard deviations (SDs) for continuous variables. We compared the K-PCAT scores between the healthcare facilities and examined the factors associated with the differences. We also evaluated correlates of the total K-PCAT score for each care facility. General linear models were used to calculate regression coefficients (βs) and standard errors (SEs) with adjustment for sociodemographic and healthcare utilization variables. Missing item values of K-PCAT were treated as zero. In sensitivity analyses, participants with any missing value for the K-PCAT items were excluded from the analyses. Statistical tests were 2-sided and $P < .05$ was considered significant. All statistical analyses were performed using SAS version 9.4 (SAS Institute Inc., Cary, NC, USA).

## Results

A total of 5,904 individuals participated in the survey (16.9% of the entire university population). After excluding respondents with missing item values on more than 80% of the K-PCAT ($n = 156$), 5,748 participants remained for analysis. Of the participants, 72.6% were students and 50.8% were women (Table 1). The mean (SD) ages were 25.7 (4.6) years for students, and 40.1 (10.3) years for faculty and staff.

### Comparison of K-PCAT scores across facilities

The K-PCAT total score was highest for the university health service (61.0 ± 15.9) and lowest for hospitals (48.1 ± 14.5), with significant differences between facilities ($P < .001$) (Table 2). Strengths in the key attributes of primary care varied across the facilities (Fig 1). The university health service received the highest scores for first contact, comprehensiveness, personalized care, and family/community orientation; community clinics for continuity; and hospitals for coordination and trust/satisfaction. Conversely, the university health service was rated lowest for continuity; community clinics for coordination, family/community orientation, and trust/satisfaction; and hospitals for first contact, comprehensiveness, continuity, and personalized care. In sensitivity analyses excluding participants with missing data on any item of the K-PCAT, results were similar: the K-PCAT total scores were 63.8(± 12.9), 57.1(± 10.5), and 51.2(± 11.0) for the university health service ($n = 4,772$), community clinics ($n = 4,879$), and hospitals ($n = 4,333$), respectively, with significant differences between facilities ($P < .001$).

### Correlates of the differences in K-PCAT total scores between facilities

We examined the factors significantly associated with the differences in the K-PCAT total scores between the facilities (Table 3). Participants who rated higher scores for the university health service than hospitals were older, students, and those with low income levels, having no regular doctor, using ambulatory care service more frequently, and spending less on medical expenses. Individuals who rated higher for community clinics than hospitals were women and those in good health, with low income levels, using ambulatory care service more frequently, and spending less on medical expenses. Correlates of higher scores for the university health service than community clinics were old age, men, students, having chronic medical conditions, having no regular doctor, and using ambulatory care services more frequently. Correlates of higher scores for the community-based primary care facilities (both the university health service and community clinics) than hospitals included good self-perceived health, low income levels, frequent ambulatory care use, and low medical

**Table 1. Sociodemographic characteristics of study participants.**

| | Total (n = 5,748) | | Student (n = 4,171) | | Faculty/staff (n = 1,577) | |
|---|---|---|---|---|---|---|
| Age, y | 29.6 | (9.3) | 25.7 | (4.6) | 40.1 | (10.3) |
| Sex | | | | | | |
| Male | 2,826 | (49.2) | 2,149 | (51.5) | 677 | (42.9) |
| Female | 2,922 | (50.8) | 2,022 | (48.5) | 900 | (57.1) |
| Income level | | | | | | |
| Low | 999 | (17.4) | 717 | (17.2) | 282 | (17.9) |
| Middle | 2,606 | (45.3) | 1,764 | (42.3) | 842 | (53.4) |
| High | 2,143 | (37.3) | 1,690 | (40.5) | 453 | (28.7) |
| Self-perceived health | | | | | | |
| Good | 2,987 | (52.0) | 2,243 | (53.8) | 744 | (47.2) |
| Fair | 2,341 | (40.7) | 1,593 | (38.2) | 748 | (47.4) |
| Poor | 420 | (7.3) | 335 | (8.0) | 85 | (5.4) |
| Comorbidity | | | | | | |
| None | 1,265 | (22.0) | 987 | (23.7) | 278 | (17.6) |
| Only acute condition | 2,684 | (46.7) | 2,019 | (48.4) | 665 | (42.2) |
| Chronic condition[a] | 1,799 | (31.3) | 1,165 | (27.9) | 634 | (40.2) |
| Regular care for chronic condition | | | | | | |
| None in the family | 3,033 | (52.8) | 2,348 | (56.3) | 685 | (43.4) |
| One-self | 283 | (4.9) | 76 | (1.8) | 207 | (13.1) |
| Family member | 2,432 | (42.3) | 1,747 | (41.9) | 685 | (43.4) |
| Medical doctor in the family | | | | | | |
| No | 5,179 | (90.1) | 3,785 | (90.8) | 1,394 | (88.4) |
| Yes | 569 | (9.9) | 386 | (9.3) | 183 | (11.6) |
| Having a regular doctor | | | | | | |
| No | 4,889 | (85.1) | 3,647 | (87.4) | 1,242 | (78.8) |
| Yes | 859 | (14.9) | 524 | (12.6) | 335 | (21.2) |
| Ambulatory care use per year, n | | | | | | |
| 0–3 | 3,402 | (59.2) | 2,544 | (61.0) | 858 | (54.4) |
| 4–6 | 1,409 | (24.5) | 984 | (23.6) | 425 | (27.0) |
| 7–12 | 609 | (10.6) | 419 | (10.1) | 190 | (12.1) |
| $\geq$13 | 328 | (5.7) | 224 | (5.4) | 104 | (6.6) |
| Medical expense per year, KRW[b] | | | | | | |
| <250,000 | 4,207 | (73.2) | 3,239 | (77.7) | 968 | (61.4) |
| 250,000–499,999 | 766 | (13.3) | 485 | (11.6) | 281 | (17.8) |
| 500,000–999,999 | 484 | (8.4) | 277 | (6.6) | 207 | (13.1) |
| $\geq$1,000,000 | 291 | (5.1) | 170 | (4.1) | 121 | (7.7) |

Values are numbers (percentages) or means ± standard deviations unless otherwise indicated.

[a] Includes hypertension, diabetes, dyslipidemia, heart disease, hyperuricemia, chronic viral hepatitis, arthritis, cancer, depression, and anxiety disorder.

[b] Out-of-pocket payment only. KRW, Korean Won. 10,000 KRW $\approx$ 8.85 USD.

expenses. Participants with low income levels rated the university health service the highest, and hospitals the lowest. We additionally investigated the correlates of the K-PCAT total scores for each facility (S1 Table). Participants with high income levels rated hospitals higher than those with low income levels. Frequent ambulatory care users perceived the university health service and community clinics more highly and hospitals lower than did less frequent users.

**Table 2. Comparison of the K-PCAT scores between different types of healthcare facilities.**

| | UHS (n = 5,454) | | Clinics (n = 5,520) | | Hospitals (n = 5,015) | | $P^a$ (UHS vs. hospitals) | $P^b$ (clinics vs. hospitals) | $P^c$ (clinics vs. UHS) |
|---|---|---|---|---|---|---|---|---|---|
| **First contact** | | | | | | | | | |
| (Utilization) When I have a new health problem, I will visit the facility first. | 3.7 | (1.1) | 3.9 | (0.9) | 2.6 | (1.1) | < .001 | < .001 | < .001 |
| (Accessibility) The facility is easy to access geographically and temporally. | 4.1 | (1.1) | 4.1 | (0.8) | 2.0 | (1.0) | < .001 | < .001 | .99 |
| (Affordability) The out-of-pocket cost is appropriate and affordable. | 4.5 | (0.8) | 3.7 | (0.8) | 2.1 | (1.0) | < .001 | < .001 | < .001 |
| Subtotal score | 12.1 | (2.7) | 11.6 | (2.2) | 6.7 | (2.5) | < .001 | < .001 | < .001 |
| **Comprehensiveness** | | | | | | | | | |
| The doctor provides comprehensive care for various health problems. | 3.3 | (1.1) | 3.0 | (0.9) | 3.7 | (1.2) | < .001 | < .001 | < .001 |
| I will visit the facility for basic health care such as periodic physical exam and blood tests. | 4.1 | (1.0) | 3.5 | (1.1) | 2.5 | (1.2) | < .001 | < .001 | < .001 |
| I will visit the facility for simple medical procedures (e.g., wound closure and dressing). | 3.9 | (1.2) | 3.8 | (1.0) | 2.3 | (1.3) | < .001 | < .001 | < .001 |
| I will consult the doctor for health counseling and education on healthy lifestyle | 3.5 | (1.3) | 2.8 | (1.2) | 2.1 | (1.1) | < .001 | < .001 | < .001 |
| I will visit the facility for a regular general health checkup before going somewhere else. | 3.7 | (1.3) | 2.8 | (1.1) | 3.2 | (1.4) | < .001 | < .001 | < .001 |
| Subtotal score | 18.0 | (4.6) | 15.4 | (4.0) | 13.4 | (4.4) | < .001 | < .001 | < .001 |
| **Continuity** | | | | | | | | | |
| The doctor knows my complete medical history and health states. | 2.6 | (1.2) | 2.8 | (1.1) | 2.6 | (1.2) | .27 | < .001 | < .001 |
| **Coordination** | | | | | | | | | |
| The doctor refers me to a specialist or special service when additional care is required. | 3.3 | (1.1) | 3.1 | (1.0) | 3.5 | (1.2) | < .001 | < .001 | < .001 |
| **Personalized care** | | | | | | | | | |
| The doctor tries to listen to and understands my words and questions well. | 3.7 | (1.0) | 3.5 | (0.9) | 3.2 | (1.1) | < .001 | < .001 | < .001 |
| The doctor provides an easy and detailed explanation of my health states and test results. | 3.7 | (1.0) | 3.5 | (0.9) | 3.4 | (1.1) | < .001 | < .001 | < .001 |
| The doctor is interested in my mental health problems as well as physical health problems. | 2.9 | (1.2) | 2.4 | (1.1) | 2.4 | (1.1) | < .001 | .008 | < .001 |
| Subtotal score | 10.3 | (2.8) | 9.3 | (2.4) | 8.9 | (2.7) | < .001 | < .001 | < .001 |
| **Family/community orientation** | | | | | | | | | |
| The doctor knows and has a concern about my family and living environment. | 2.3 | (1.1) | 2.3 | (1.1) | 2.1 | (1.1) | < .001 | < .001 | .73 |
| The doctor is active in promoting the community health (health courses, home visits, etc.) | 3.4 | (1.2) | 2.6 | (1.2) | 2.8 | (1.2) | < .001 | < .001 | < .001 |
| The facility surveys and reflects patients' opinions to provide better health care. | 3.6 | (1.2) | 2.4 | (1.1) | 2.9 | (1.2) | < .001 | < .001 | < .001 |
| Subtotal score | 9.1 | (2.9) | 7.2 | (2.9) | 7.6 | (2.8) | < .001 | < .001 | < .001 |
| **Trust/satisfaction** | | | | | | | | | |
| I can trust the doctor's decisions on treatment. | 3.7 | (0.9) | 3.5 | (0.8) | 4.1 | (0.9) | < .001 | < .001 | < .001 |
| Overall, the health care service provided is satisfactory. | 3.9 | (0.9) | 3.6 | (0.8) | 3.7 | (0.9) | < .001 | < .001 | < .001 |
| Subtotal score | 7.6 | (1.8) | 7.0 | (1.6) | 7.7 | (1.6) | < .001 | < .001 | < .001 |
| **Total score** | 61.0 | (15.9) | 55.1 | (12.9) | 48.1 | (14.5) | < .001 | < .001 | < .001 |

K-PCAT = Korean Primary Care Assessment Tool, UHS = university health service. Each item score ranges from 1 (low) to 5 (high).

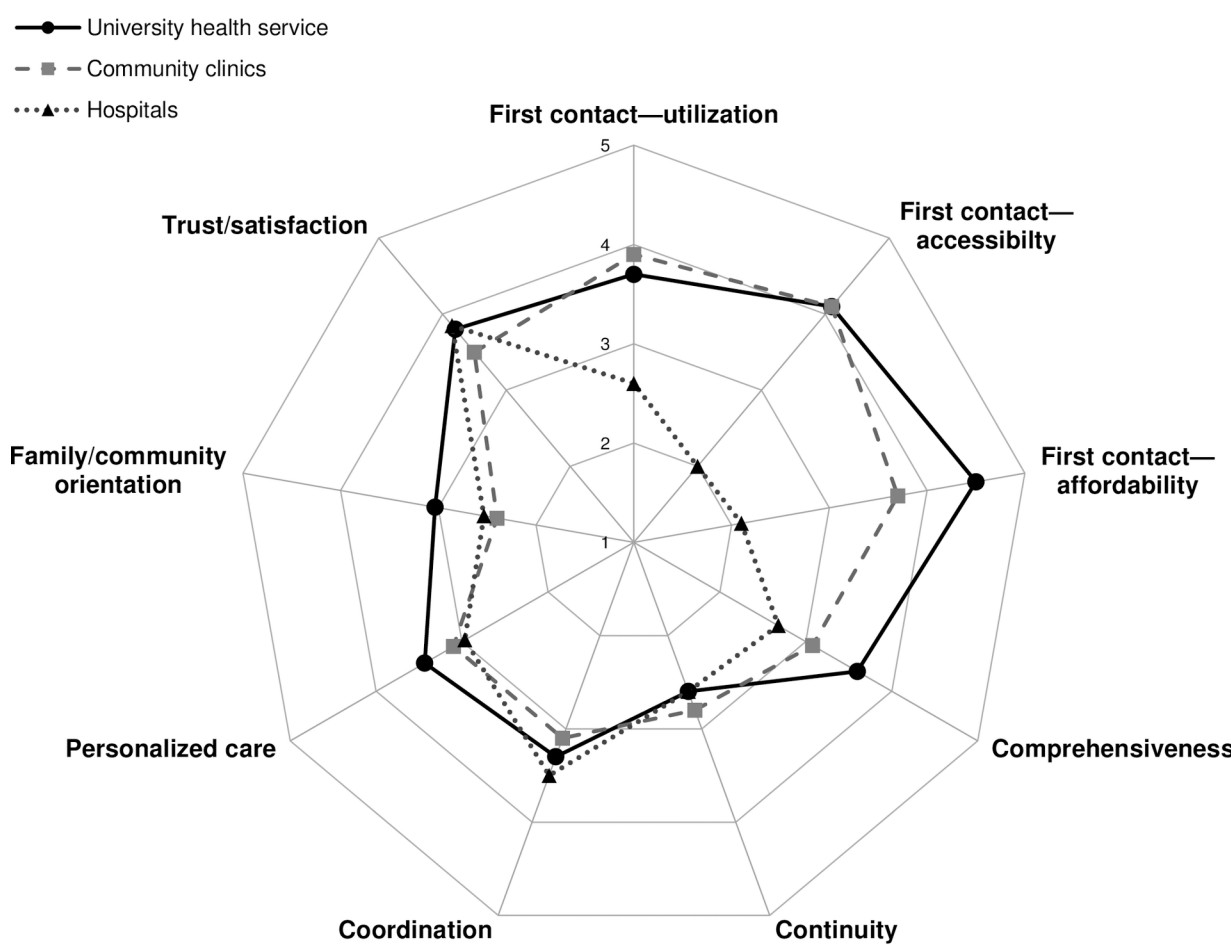

**Fig 1. Comparison of patient-perceived primary care qualities across healthcare facilities by domains of K-PCAT.** K-PCAT = Korean Primary Care Assessment Tool. Each domain score was calculated as the mean of subordinate item scores. Each item score ranged from 1 (low) to 5 (high).

## Discussion

In this cross-sectional study, the user-perceived qualities of primary care were higher for the community-based primary care facilities than hospitals. Although strengths in the key attributes of primary care varied across the facilities, the university health service received the highest score in overall performance, and hospitals received the lowest score.

### Perspectives of primary care

The concept of primary care is defined in Korea as "a healthcare service first encountered by people. Primary care physicians resolve most of the common health needs of the population, and see patients in the context of family and community with continuous doctor-patient relationships, coordinating healthcare resources appropriately."[33] The perspective of primary care is built on the principles of clinical efficiency, cost-effectiveness, and equitable care for the population.[34] Previous studies demonstrated that specialist-dominated health systems enforce inequity in access, whereas primary-care-oriented systems tend to be more pro-poor, equitable, and accessible.[6,10,11,35] Broadening access to primary care can reduce demand for expensive, specialist-led hospital care. In our study, individuals with low income levels,

**Table 3. Factors associated with the differences in K-PCAT total scores between healthcare facilities.**

| | UHS vs. hospital (n = 4,817) | | | | Clinic vs. hospital (n = 4,969) | | | | UHS vs. clinic (n = 5,230) | | | |
|---|---|---|---|---|---|---|---|---|---|---|---|---|
| | β | SE | P | $P_{trend}$ | β | SE | P | $P_{trend}$ | β | SE | P | $P_{trend}$ |
| Age | 0.18 | 0.04 | **< .001** | | 0.02 | 0.03 | .647 | | 0.14 | 0.03 | **< .001** | |
| Sex | | | | | | | | | | | | |
| Male | (ref) | – | | | (ref) | – | | | (ref) | – | | |
| Female | -0.52 | 0.50 | .327 | | 2.28 | 0.42 | **< .001** | | -2.23 | 0.43 | **< .001** | |
| Job | | | | | | | | | | | | |
| Student | (ref) | – | | | (ref) | – | | | (ref) | – | | |
| Staff | -5.15 | 0.80 | **< .001** | | 0.02 | 0.66 | .974 | | -5.94 | 0.70 | **< .001** | |
| Faculty | -5.13 | 1.09 | **< .001** | | -1.44 | 0.90 | .109 | | -3.18 | 0.97 | **.001** | |
| Income level | | | | **< .001** | | | | **< .001** | | | | .376 |
| Low | (ref) | – | | | (ref) | – | | | (ref) | – | | |
| Middle | -1.56 | 0.69 | **.023** | | -2.08 | 0.58 | **< .001** | | 0.23 | 0.59 | .704 | |
| High | -2.70 | 0.73 | **< .001** | | -2.62 | 0.61 | **< .001** | | -0.40 | 0.63 | .522 | |
| Self-perceived health | | | | .206 | | | | **.011** | | | | .304 |
| Good | (ref) | – | | | (ref) | – | | | (ref) | – | | |
| Fair | -1.15 | 0.54 | **.033** | | -0.88 | 0.45 | .051 | | 0.11 | 0.47 | .812 | |
| Poor | -0.08 | 0.99 | .938 | | -1.81 | 0.84 | **.031** | | 1.20 | 0.88 | .173 | |
| Comorbidity | | | | | | | | | | | | |
| None | (ref) | – | | | (ref) | – | | | (ref) | – | | |
| Acute disease only | 0.04 | 0.66 | .954 | | 0.43 | 0.55 | .439 | | -0.06 | 0.57 | .916 | |
| Chronic condition[a] | 1.02 | 0.75 | .173 | | -0.13 | 0.63 | .836 | | 1.28 | 0.65 | **.049** | |
| Hospital visit for chronic condition | | | | | | | | | | | | |
| No | (ref) | – | | | (ref) | – | | | (ref) | – | | |
| Yes | 0.00 | 0.49 | .994 | | 0.44 | 0.41 | .284 | | -0.17 | 0.43 | .682 | |
| Medical doctor in the family | | | | | | | | | | | | |
| No | (ref) | – | | | (ref) | – | | | (ref) | – | | |
| Yes | -1.41 | 0.84 | .092 | | -0.40 | 0.70 | .570 | | -1.48 | 0.74 | **.045** | |
| Having a regular doctor | | | | | | | | | | | | |
| No | (ref) | – | | | (ref) | – | | | (ref) | – | | |
| Yes | -2.46 | 0.71 | **.001** | | 0.33 | 0.59 | .583 | | -2.86 | 0.63 | **< .001** | |
| Ambulatory care visit per year, n | | | | **< .001** | | | | **< .001** | | | | **.024** |
| 0–3 | (ref) | – | | | (ref) | – | | | (ref) | – | | |
| 4–6 | 1.60 | 0.61 | **.009** | | 1.00 | 0.51 | .051 | | 0.28 | 0.53 | .598 | |
| 7–12 | 3.91 | 0.87 | **< .001** | | 1.58 | 0.73 | **.031** | | 1.97 | 0.77 | **.010** | |
| ≥13 | 6.22 | 1.19 | **< .001** | | 4.54 | 1.00 | **< .001** | | 1.45 | 1.05 | .166 | |
| Medical expense per year, KRW[b] | | | | **< .001** | | | | **< .001** | | | | .248 |
| <250,000 | (ref) | – | | | (ref) | – | | | (ref) | – | | |
| 250,000–499,999 | -3.62 | 0.75 | **< .001** | | -2.11 | 0.63 | **.001** | | -1.43 | 0.67 | **.032** | |
| 500,000–999,999 | -3.69 | 0.93 | **< .001** | | -3.03 | 0.78 | **< .001** | | -0.44 | 0.83 | .595 | |
| ≥1,000,000 | -5.35 | 1.21 | **< .001** | | -3.80 | 1.01 | **< .001** | | -0.86 | 1.08 | .426 | |

K-PCAT = Korean Primary Care Assessment Tool, β = regression coefficient, SE = standard errors, UHS = university health service.

Statistically significant results are marked in bold.

[a] Includes hypertension, diabetes, dyslipidemia, heart disease, hyperuricemia, chronic viral hepatitis, arthritis, cancer, depression, and anxiety disorder.

[b] Out-of-pocket payment only. KRW, Korean Won. 10,000 KRW ≈ 8.85 USD.

using ambulatory care services more frequently, but spending less on medical expenses rated primary care facilities higher than hospitals. These results suggest that providing high-quality

primary care can expand healthcare utilization for the low-income population at lower cost. In particular, the university health service was better assessed among students, individuals with chronic medical conditions, and those without a regular doctor. Therefore, it may fulfill the unmet needs of these vulnerable groups in a university, enhance their accessibility, expand service utilization, and reduce health inequalities.

## A setting-based, people-centered primary care model

Setting-based (e.g., schools and workplaces) health promotion has been emphasized by the WHO.[36] People-centeredness is one of the core principles of primary care: through delivery points embedded in communities, health systems can better respond to people's needs.[2,7] Previous studies indicated that a higher level of population health was achieved when primary care providers served a well-defined population in a designated area, knowing their target population better.[4,30] In the present study, the university health service received the highest user-perceived score and had strengths in many attributes of primary care. A university health service is usually the most accessible primary care point on campus; care providers on campus best apprehend the university's health issues, resources, and physical and sociocultural environments within a campus. Therefore, it can serve as an excellent model of setting-based, people-centered primary care. The main financing of the university health service, where this study was conducted, is from the university funding with some revenue through reimbursement by the National Health Insurance and out-of-pocket payments. Thus, it has a mixed financing of public, self-governing, and non-profit models. Similar to our results, an alternative non-profit primary care service was evaluated more highly in the performance of primary care than public or private/for-profit facilities in Korea and other countries [34,35]

Continuity of care, a sustained partnership between patients and physicians, is a key element of high-quality primary care.[37] A doctor knowing a patient's history and values can provide more suitable care.[38] Many previous studies indicated that continuity of care can improve patient adherence and self-management, improved health outcomes, and lower unnecessary healthcare utilization and costs.[38–40] We found that the university health service needs to be improved in continuity of care. Because this weakness might be attributed to the properties of group practice, it could be overcome by establishing a regular doctor system.

## Primary care services of community clinics

Community clinics received higher user assessments than hospitals; however, their score was lower than that of the university health service. Specifically, community clinics had the lowest scores for care coordination, family/community orientation, and trust/satisfaction. These findings can be explained by the following reasons, mostly linked to the fundamental problems of the Korean health system. First, primary care services of community clinics are heterogeneous. Many of community doctors in Korea are specialists and sub-specialties rather than general physicians (e.g., family doctors).[16,23,24] Since the range and qualifications of primary care providers have never been explicitly defined at the national level, all medical doctors can run a private clinic and practice as the first-contact point, regardless of their specialties.[16,20,22] Prior studies revealed that most specialists had no correct understanding of the roles of primary care and tended to underuse preventive care, focusing on curative and specialty care, [7,16,23] whereas general physicians were more likely to provide comprehensive and family/community-oriented care.[6,20,35]

Second, the private sector, aiming to maximize profits, dominates the health system in the absence of clear distinctions between the roles of hospitals and community clinics. Despite the inherent characteristic of health services as public commodities, these circumstances have

shaped the health system to resemble a laissez-faire market, consequently leading to excessive competition between care providers.[16,34] Health services, including primary care, are highly fragmented and overlapping with little coordination between providers, even in the National Health Insurance system.[21–24] A survey of 466 general internists reported that coordination and comprehensiveness were most vulnerable attributes of primary care in Korea; one third of the physicians were reluctant to refer their patients for fear of losing them.[18]

Third, the fee-for-service payment system, a dominant reimbursement schedule in Korea, is unfavorable for achieving the key attributes of primary care (e.g., comprehensiveness, coordination).[16,18] As most services for patient education are not reimbursed, there is little financial incentive for doctors to focus on disease prevention and health promotion.[18,22]

Lastly, an overwhelming majority (around 94%) of community clinics are run by solo practitioners.[17] Hence, they have strength in continuity of care as found in our results, but tend to be limited in manpower and infrastructure to ensure quality of care.[22]

## Hospital outpatient care

Hospital outpatient services were rated lowest in the performance of primary care of all facilities with many important primary care attributes unfulfilled. Since the original mission of hospitals is the provision of specialty care for complicated or rare cases, not comprehensive and integrated primary care, our results seem to be inevitable. Although hospitals were highly assessed in care coordination, this might imply that hospital patients tended to be easily referred from a specialist to other specialists within the same hospital once patients entered the hospital system,[17] possibly leading to fragmented care and over-utilization of specialty care.

Globally, the importance of providing healthcare at the right time and place, without compromising on care quality, is being emphasized.[3] Unnecessary utilization of hospital outpatient care is one of the main contributors of escalating healthcare costs.[25] Substituting hospital outpatient care with primary care is recommended as a solution to accomplish efficient and cost-effective healthcare delivery.[1,3] Evidence supports this perspective, indicating shorter waiting lists, shorter clinic waiting times, and higher patient satisfaction after the substitution interventions.[25]

Nonetheless, we found that the levels of trust and user satisfaction were highest for hospitals. This user view might be partly linked to the belief that good facilities and equipment of large hospitals guarantee good qualities of care.[13,23] Similar user misperception and distorted health-seeking behaviors were also reported in China; community individuals preferred to seek care in second or tertiary hospitals rather than primary care facilities, although the latter provide more accessible and affordable care. An important driver for this health-seeking behavior was trust in doctors in tertiary hospitals.[13] Therefore, efforts to raise public trust in primary care physicians and facilities are essential.

## Strengths and limitations

Our study has several strengths. We enrolled a large population in a university setting. To the best of our knowledge, studies of primary care in a university setting are rare worldwide. Only 2 universities in Korea, including the university where this study was conducted, have accredited primary care services on campus covered by the National Health Insurance System. Because Korean citizens can freely choose and visit multiple healthcare facilities without regulation,[16] we were able to compare the real-world patient experiences of primary care services in the community population. To obtain information that was as unbiased as possible, the questionnaire was self-administered and anonymity was ensured. Relatively homogeneous

participants exposed to similar geographic and sociocultural environments were sampled, which might have reduced the possibility of unmeasured confounding.

This study has following limitations. First, the cross-sectional design of the study limits causal and temporal inferences. Second, we evaluated user experiences and perception rather than health outcomes. Third, because we did not specify particular clinics, hospitals, and specialties, heterogeneity within the comparison groups might have affected our results. However, it might also have helped us to aggregate various user experiences in the real world. Fourth, we were not able to rule out sampling bias. Because our sample was recruited from a university, this might have influenced the positive results for the university service. Although we did not restrict our sample to the users of the university health service to decrease such bias, our sample size (16.9% of the study population) and the non-random sampling method could have limited its representativeness. Fifth, covariate information was obtained by participants' self-report, thus we cannot exclude the possibility of misclassification. Lastly, our findings, derived from a single university, may not be generalizable to the entire Korean population or applicable to different health systems.

## Conclusions

The user-perceived primary care quality was higher for community-based primary care facilities than hospitals. The university health service received the highest score in the performance of primary care, suggesting that setting-based, patient-centered primary care may serve as an effective model for restructuring the Korean health system. Since primary care is based on community participation and collaboration between different sectors of society,[6] efforts to raise the awareness of the concept and benefits of primary care among health professionals, policy makers, and the public will be necessary.

## Supporting information

**S1 Data. Data underlying the presented results.**
(SAS7BDAT)

**S1 File. Questionnaire used in the study (English).**
(DOCX)

**S2 File. Questionnaire used in the study (Korean).**
(DOCX)

**S1 Table. Factors associated with K-PCAT total scores for each healthcare facility.**
K-PCAT = Korean Primary Care Assessment Tool, β = regression coefficient, SE = standard errors, UHS = university health service Statistically significant results are marked in bold. [a] Includes hypertension, diabetes, dyslipidemia, heart disease, hyperuricemia, chronic viral hepatitis, arthritis, cancer, depression, and anxiety disorder [b] Out-of-pocket payment only. KRW, Korean Won. 10,000 KRW ≈ 8.85 USD.
(DOCX)

## Author Contributions

**Conceptualization:** Yongjung Cho, Heeyoung Chung, Hyundeok Joo, Hee-Kyung Joh, Ji Won Kim, Jong-Koo Lee.

**Data curation:** Yongjung Cho, Heeyoung Chung, Hyung Jun Park, Hee-Kyung Joh.

**Formal analysis:** Hee-Kyung Joh.

**Funding acquisition:** Hee-Kyung Joh, Ji Won Kim.

**Investigation:** Heeyoung Chung, Hyundeok Joo, Hee-Kyung Joh, Jong-Koo Lee.

**Methodology:** Yongjung Cho, Hee-Kyung Joh, Jong-Koo Lee.

**Project administration:** Hee-Kyung Joh, Jong-Koo Lee.

**Resources:** Hyundeok Joo, Hee-Kyung Joh, Jong-Koo Lee.

**Supervision:** Hee-Kyung Joh, Jong-Koo Lee.

**Validation:** Hee-Kyung Joh.

**Visualization:** Hee-Kyung Joh.

**Writing – original draft:** Yongjung Cho, Heeyoung Chung, Hee-Kyung Joh.

**Writing – review & editing:** Yongjung Cho, Hyung Jun Park, Hee-Kyung Joh, Jong-Koo Lee.

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
