## [Decision Letter · Decision Letter 0]

23 Dec 2019

PONE-D-19-31223

Comparison of patient perceptions of primary care quality across healthcare facilities in Korea: a cross-sectional study

PLOS ONE

Dear Dr Joh,

Thank you for submitting your manuscript to PLOS ONE. After careful consideration, we feel that it has merit but does not fully meet PLOS ONE’s publication criteria as it currently stands. Therefore, we invite you to submit a revised version of the manuscript that addresses the points raised during the review process.

In order to provide a more complete information to our readers on the topic, we would like to emphasize the importance to cross referencing very recent material on the same topic published in "PLoS ONE ". Therefore, it would be highly appreciated if you would check the contents published in the last two years of "PLoS ONE" (https://journals.plos.org/plosone/) and add all material relevant to your article to the reference list.

We would appreciate receiving your revised manuscript by Feb 06 2020 11:59PM. To enhance the reproducibility of your results, we recommend that if applicable you deposit your laboratory protocols in protocols.io, where a protocol can be assigned its own identifier (DOI) such that it can be cited independently in the future. For instructions see: http://journals.plos.org/plosone/s/submission-guidelines#loc-laboratory-protocols

We look forward to receiving your revised manuscript.

Kind regards,

Wen-Jun Tu

Academic Editor

PLOS ONE

Journal Requirements:

Reviewers' comments:

Reviewer's Responses to Questions

**Comments to the Author**

1. Is the manuscript technically sound, and do the data support the conclusions?

Reviewer #1: Partly

2. Has the statistical analysis been performed appropriately and rigorously? 

Reviewer #1: Yes

3. Have the authors made all data underlying the findings in their manuscript fully available?

Reviewer #1: Yes

4. Is the manuscript presented in an intelligible fashion and written in standard English?

Reviewer #1: Yes

5. Review Comments to the Author

Reviewer #1: The manuscript is of great relevance to public health. But I believe the discussion could be more robust, given the data presented.

In the discussion the main results are presented unnecessarily as they are all available in the "results" field. A large number of articles used in the discussion have more than 10 years of publication (33.0%). The inclusion of more current articles is necessary.

I suggest discussing about the final sample size, 16.9% of the sample recruited. In addition to this issue, the sample was restricted to university users. Could these questions have influenced the results? Was the final sample size representative? The fact that they are all from university may have influenced the positive result for the service offered in this establishment? Can these questions be a limitation of the study?

I also suggest reviewing the formatting of the text and reviewing the writing of very long paragraphs.

6. PLOS authors have the option to publish the peer review history of their article (what does this mean?). If published, this will include your full peer review and any attached files.

Reviewer #1: Yes: Fernanda Piana Santos Lima de Oliveira

---

## [Author Response · Author response to Decision Letter 0]

6 Feb 2020

Dear Editor, 

We deeply appreciate the opportunity to revise and re-submit our manuscript, titled “Comparison of patient perceptions of primary care quality across healthcare facilities in Korea: a cross-sectional study” to PLOS ONE. The reviewers' comments and suggestions were highly valuable in improving the quality of our manuscript. Below, we have addressed the editor’s and reviewers’ concerns in a point-by-point manner. We hope that the revised manuscript meets with your approval for publication.

Again, thank you very much for your thorough review of our manuscript.

Sincerely,

Hee-Kyung Joh 

 

Responses to academic editor:

 Per the editor’s comment, we have modified our manuscript and file names to meet the PLOS ONE’s style requirements. 

 Per the editor’s comment, we have included a copy of the questionnaire that we used in our study as Supporting Information (S1 and S2 Files) in both the original language and English.

 Per the editor’s comment, we have included captions for the Supporting Information files at the end of our manuscript (page 27), and updated in-text citations to match accordingly (page 8, line 103; page 13, line 193).

 We acknowledge that there are no restrictions on sharing the de-identified data set that we have uploaded as a Supporting Information file (S1 Data). We have indicated this information in the revised cover letter. 

Responses to Reviewer #1

1. The manuscript is of great relevance to public health. But I believe the discussion could be more robust, given the data presented. 

 We appreciate the reviewer’s comment. Per the reviewer’s suggestion, we have revised our Discussion section to make it more solid and based on our study results. We also have reorganized the paragraphs and have added more discussions on outpatient hospital care. The following paragraphs are one example: “Globally, the importance of providing healthcare at the right time and place, without compromising on care quality, is being emphasized.[3] Unnecessary utilization of hospital outpatient care is one of the main contributors of escalating healthcare costs.[25] Substituting hospital outpatient care with primary care is recommended as a solution to accomplish efficient and cost-effective healthcare delivery.[1,3] Evidence supports this perspective, indicating shorter waiting lists, shorter clinic waiting times, and higher patient satisfaction after the substitution interventions.[25]

Nonetheless, we found that the levels of trust and user satisfaction were highest for hospitals. This user view might be partly linked to the belief that good facilities and equipment of large hospitals guarantee good qualities of care.[13,23] Similar user misperception and distorted health-seeking behaviors were also reported in China; community individuals preferred to seek care in second or tertiary hospitals rather than primary care facilities, although the latter provide more accessible and affordable care. An important driver for this health-seeking behavior was trust in doctors in tertiary hospitals.[13] Therefore, efforts to raise public trust in primary care physicians and facilities are essential.” (Manuscript file, page 19–20, line 292–306)

Please refer to the ‘Manuscript’ file (page 16–21) and ‘Revised Manuscript with Track Changes’ file (page 16–22) to check the entire revision of the Discussion section. We have highlighted in yellow for the newly added parts in the Revised Manuscript with Track Changes file. 

2. In the discussion the main results are presented unnecessarily as they are all available in the "results" field.

 Per the reviewer’s comment, we have summarized the main results as concise as possible in the revised version. In addition, we have moved the following sentences to the Method section: “A university health service is a primary care model on campus dealing with medical care and health promotion of university students and employees. The university health service in this university implements a wide range of activities: primary medical consultation for acute and chronic diseases, mental health care, communicable disease prevention, immunizations, health screening, and health education. Family physicians play the central role in these activities, partly supported by some specialty care.”(Manuscript file, page 7, line 94–99)

3. A large number of articles used in the discussion have more than 10 years of publication (33.0%). The inclusion of more current articles is necessary.

 Per the reviewer’s comment, we have additionally reviewed more recent studies and articles. In the revised version, 16 papers have been newly added, and 13 of them were published within 10 years. Overall, 28 of 40 (70%) references were published within 10 years, and 11 references within 2 years. 

4. I suggest discussing about the final sample size, 16.9% of the sample recruited. In addition to this issue, the sample was restricted to university users. Could these questions have influenced the results? Was the final sample size representative? The fact that they are all from university may have influenced the positive result for the service offered in this establishment? Can these questions be a limitation of the study?

 We agree with the reviewer’s point. Per the reviewer’s suggestion, we have added this point in the limitations as follows: “Fourth, we cannot rule out sampling bias. Because our sample was recruited from a university, this might have influenced the positive results for the university service. Although we did not restrict our sample to the users of the university health service to decrease such bias, our sample size (16.9% of the study population) and the non-random sampling method could have limited its representativeness.” (Manuscript file, page 21, line 323–327)

5. I also suggest reviewing the formatting of the text and reviewing the writing of very long paragraphs.

 We appreciate the reviewer’s comment. Per the reviewer’s suggestion, the Discussion section was re-formatted. We have re-organized paragraphs to be shorter, using several subtitles, to enhance readers’ readability. Also, long sentences were trimmed as concisely as possible. Please refer to the ‘Manuscript’ file (page 16–21) and ‘Revised Manuscript with Track Changes’ file (page 16–22) to check the entire revision of the Discussion section.

---

## [Decision Letter · Decision Letter 1]

20 Feb 2020

Comparison of patient perceptions of primary care quality across healthcare facilities in Korea: a cross-sectional study

PONE-D-19-31223R1

Dear Dr. Joh,

We are pleased to inform you that your manuscript has been judged scientifically suitable for publication and will be formally accepted for publication once it complies with all outstanding technical requirements.

With kind regards,

Wen-Jun Tu

Academic Editor

PLOS ONE

Additional Editor Comments (optional):

Reviewers' comments:

Reviewer's Responses to Questions

**Comments to the Author**

1. If the authors have adequately addressed your comments raised in a previous round of review and you feel that this manuscript is now acceptable for publication, you may indicate that here to bypass the “Comments to the Author” section, enter your conflict of interest statement in the “Confidential to Editor” section, and submit your "Accept" recommendation.

Reviewer #1: (No Response)

2. Is the manuscript technically sound, and do the data support the conclusions?

Reviewer #1: (No Response)

3. Has the statistical analysis been performed appropriately and rigorously? 

Reviewer #1: (No Response)

4. Have the authors made all data underlying the findings in their manuscript fully available?

Reviewer #1: (No Response)

5. Is the manuscript presented in an intelligible fashion and written in standard English?

Reviewer #1: (No Response)

6. Review Comments to the Author

Reviewer #1: (No Response)

7. PLOS authors have the option to publish the peer review history of their article (what does this mean?). If published, this will include your full peer review and any attached files.

Reviewer #1: Yes: Fernanda Piana Santos Lima de Oliveira

---

## [Editor Report · Acceptance letter]

24 Feb 2020

PONE-D-19-31223R1 

Comparison of patient perceptions of primary care quality across healthcare facilities in Korea: a cross-sectional study 

Dear Dr. Joh:

I am pleased to inform you that your manuscript has been deemed suitable for publication in PLOS ONE. Congratulations! Your manuscript is now with our production department. 

With kind regards,

on behalf of

Dr. Wen-Jun Tu 

Academic Editor

PLOS ONE